# A Pilot Study: Contrasting Genomic Profiles of Lung Adenocarcinoma Between Patients of European and Latin American Ancestry

**DOI:** 10.3390/ijms26104865

**Published:** 2025-05-19

**Authors:** Bertha Rueda-Zarazua, Humberto Gutiérrez, Humberto García-Ortiz, Lorena Orozco, Gustavo Ramírez-Martínez, Luis Jiménez-Alvarez, Francina V. Bolaños-Morales, Joaquín Zuñiga, Federico Ávila-Moreno, Jorge Melendez-Zajgla

**Affiliations:** 1Posgrado en Ciencias Biológicas, Universidad Nacional Autónoma de México, Mexico City 04510, Mexico; bgrueda@inmegen.edu.mx; 2Laboratorio de Genómica Funcional del Cáncer, Instituto Nacional de Medicina Genómica, Mexico City 14610, Mexico; 3Instituto Nacional de Medicina Genómica, Mexico City 14610, Mexico; hgutierrez@inmegen.gob.mx; 4Laboratorio de Inmunogenómica y Enfermedades Metabólicas, Instituto Nacional de Medicina Genómica, Mexico City 14610, Mexico; hgarcia@inmegen.gob.mx (H.G.-O.); lorozco@inmegen.gob.mx (L.O.); 5Laboratorio de Inmunología y Genética, Instituto Nacional de Enfermedades Respiratorias, Mexico City 14080, Mexico; grmunam@yahoo.com.mx (G.R.-M.); jimenez.alvarez.l@gmail.com (L.J.-A.); joaquin.zuniga@iner.gob.mx (J.Z.); 6Subdirección de Cirugía, Instituto Nacional de Enfermedades Respiratorias, Mexico City 14080, Mexico; 7Tecnologico de Monterrey, Escuela de Medicina y Ciencias de la Salud, Mexico City 14380, Mexico; 8Lung Diseases and Functional Epigenomics Laboratory (LUDIFE), Biomedicine Research Unit (UBIMED), Facultad de Estudios Superiores-Iztacala, Universidad Nacional Autónoma de México, Tlalnepantla 54090, Mexico; f.avila@unam.mx; 9Research Tower, Subdirección de Investigación Básica, Instituto Nacional de Cancerología (INCan), Mexico City 14080, Mexico

**Keywords:** NSCLC, lung adenocarcinoma, genomic profile, Mexico, exome

## Abstract

Lung cancer remains as the leading cause of cancer mortality worldwide. However, while current evidence suggests the existence of genomic differences between populations, indicating different risk factors associated with population-level genetic backgrounds, most studies have concentrated on populations of European ancestry, and more research is needed on non-European populations. We analyzed whole-exome sequencing data from 25 Mexican lung adenocarcinoma patients and compared them with a TCGA-PanCancer cohort enriched with patients of European ancestry as reference. Clinically relevant germline variants in cancer susceptibility genes are more frequent in our cohort (32% vs. 6.4%) than in the reference. Several mutational signatures (SBS32, SBS85, SBS12, SBS19) occurred at significantly higher frequencies in the Mexican cohort compared to the reference (*p* < 0.0001). Interestingly, the smoking-associated signature SBS4, present in 67.6% of smokers in the reference cohort, was absent in smoking Mexican patients (*p* < 0.01656). Somatic variant frequencies in SLC36A4 (20%; *p* < 0.00002), AP1S1 (8%; *p* < 0.00002), and TP53 (16%; *p* = 0.00005) showed significant differences from the European reference cohort. We demonstrate that all these observed biases were independent of the sample size. This study uncovers distinct genomic biases in lung cancer carcinogenesis in this population, compared to a European ancestry reference population, suggesting implications for precision medicine strategies in Latin American populations.

## 1. Introduction

Lung cancer continues to be the most common and deadly type of cancer worldwide. Lung adenocarcinoma (LUAD) stands as the most prevalent subtype of lung cancer in the world [1,2]. The genomic landscape of lung adenocarcinoma has been extensively examined, revealing notable differences across various populations. Among the most scrutinized aspects is the variation in the frequency of EGFR mutations. These mutations are predominantly associated with individuals of Asian ancestry, females, and nonsmokers, and are often associated with a more favorable prognosis [3,4]. In contrast, KRAS is frequently mutated in populations of European ancestry and is mainly associated with smoking. In Latin American countries, observed frequencies of mutations in both genes occupy an intermediate position between those of the previously mentioned populations. This has sparked vigorous debates, pointing towards the influence of genetic backgrounds or the impact of distinct risk factors tied to each population as cultural and environmental variations [5,6]. This pattern underscores the importance of considering genetic diversity in the diagnosis and treatment of lung adenocarcinoma.

The underrepresentation of certain populations in large-scale genomic studies, particularly those from Latin American countries, often leads to the unwarranted overgeneralization of findings derived predominantly from European or Asian cohorts. These generalized conclusions can have profound implications for clinical decision-making, potentially misguiding the understanding of disease progression and therapeutic responses. Latin American populations are uniquely characterized by their rich genetic diversity, shaped by complex migration patterns, cultural heterogeneity, and distinct sociodemographic factors. This lack of representation not only delays the equitable distribution of benefits from medical research to these populations but also impedes the advancement of precision medicine approaches tailored to their specific genetic and environmental contexts. Addressing this gap is crucial to fostering inclusive research and ensuring that genomic medicine serves global health needs effectively [7,8,9].

Research in Mexican populations has identified several clinically significant genetic variants and has also highlighted specific risk factors for lung cancer, such as exposure to wood smoke, which significantly impacts a large segment of the population [10,11]. However, previous studies have only included a limited number of genes of interest, and there is still much to be understood about the peculiarities that may exist within this population. This study aims to identify, using whole exome sequencing, differences in the mutational profiles of lung adenocarcinoma among Mexican patients, compared to those typically observed in European ancestry patients.

## 2. Results

### 2.1. Study Population

In the present study, 25 patients with confirmed diagnosis of lung adenocarcinoma were included. All patients were from the central region of Mexico and were diagnosed at the National Institute of Respiratory Diseases in Mexico City. Denominators correspond to the patient’s clinical information where available. The study included 11 males and 14 females. The majority of the patients (9 out of 14, 64%) were at stage IV and the mean age was 58.8 ± 14.5 years. A total of 66% (8 out of 12) of the participants reported a family history of cancer. Among the risk factors, 60% (10 out of 17) reported a history of smoking, while 28% (4 out of 14) reported exposure to wood smoke as a risk factor, all of whom were women. Three patients reported exposure to pesticides, fine clay dust, and asbestos as other risk factors (Figure 1A).

In order to determine the ancestry of our patients, we employed SNPs masked from three distinct base populations: African (YRI), northern European (CEU), and indigenous Mexican from the García-Ortiz’s previous study [12]. Our cohort’s primary contribution is indigenous Mexican ancestry, accounting for 53.27% of the total. This is followed by European ancestry at 42.38% and African ancestry at 4.8% (Figure 1B). The population structure of our cohort is comparable to that observed in a group of open population controls of Mexicans from central Mexico, where contributions of Indigenous Mexican, European, and African ancestry were observed to be 69.52%, 25.56%, and 2.68%, respectively, using the same ancestry references [13]. All patients have some proportion of indigenous Mexican ancestry.

**Figure 1 ijms-26-04865-f001:**
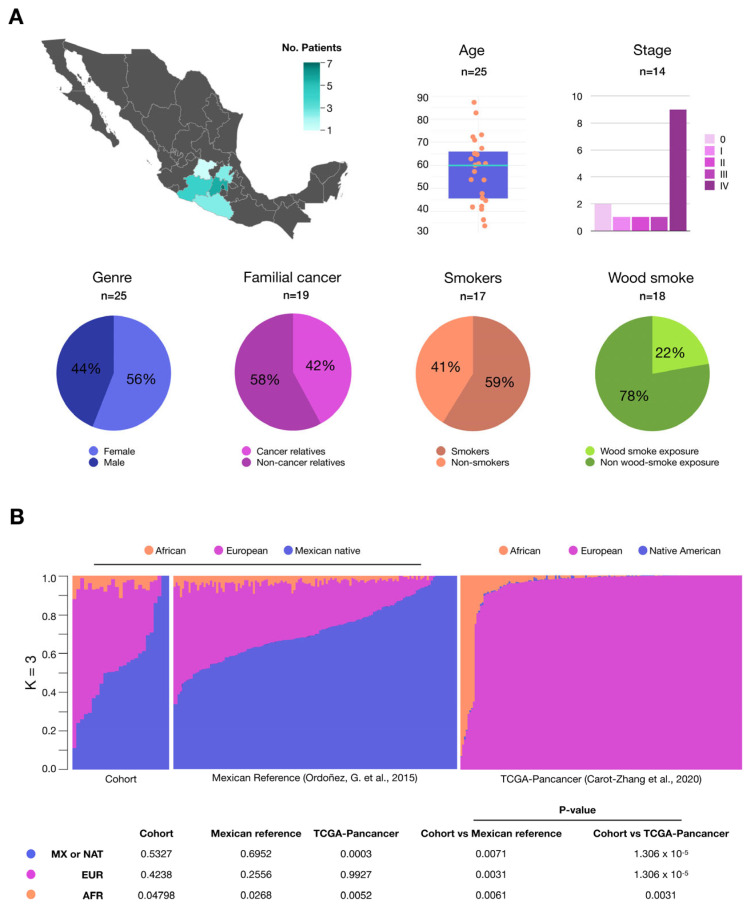
General description of the cohort of Mexican patients with lung adenocarcinoma used in this study. (**A**) Clinical features and risk factors of our Mexican patients. (**B**) Population ancestry structure of our cohort of Mexican mestizos contrasted with an open population control group from central [13] Mexico and the TCGA-PanCancer reference cohort. The Mexican cohorts used genetic markers of European, African, and indigenous Mexican ancestry, while the TCGA-PanCancer cohort [14] used markers of Native American ancestry, which is closely related to indigenous Mexican populations. Estimated individual proportions of ancestral backgrounds revealed a significant contribution from Native Mexican ancestry in the Mexican cohorts. In contrast, the TCGA reference cohort exhibited a predominant contribution from European ancestry. Comparison between cohorts was performed with one-sample Wilcoxon tests using medians.

We use the TCGA-PanCancer cohort (syn1759475) as a reference, which predominantly consists of patients of European ancestry, reflecting the most extensively studied population. This cohort, comprising over 90% European ancestry (Figure 1B) [14], is approximately 10 times larger than our own and is well-documented. It includes a higher proportion of smokers (85% vs. 59%) and exhibits a more uniform distribution of disease stages, which may introduce slight biases in smoking-related and stage-related features. Nevertheless, these differences enable us to effectively compare and identify variations in the genomic landscape.

### 2.2. Germline Variants

To evaluate potential genetic susceptibility factors, we identified germline gene variants present in our cohort. We only considered variants that were part of the list of 152 genes previously associated with cancer susceptibility [15] and classified as “Pathogenic” or “Likely pathogenic” according to the American College of Medical Genetics and Genomics (ACMG) and the Association for Molecular Pathology (AMP) recommendations [16]. Furthermore, we explored Uncertain Significance variants using the results from pathogenicity prediction tools (see Methods). A total of 20 variants were identified (Appendix A); some of them affect Homologous Recombination-related genes (Figure 2A).

We use the study by Huang et al., 2018 [15] as a reference cohort because they used the data from patients of the PanCancer cohort with a method similar to ours for the identification of germline variants. We found that 32% of patients had pathogenic or likely pathogenic germline variants, compared to 6.4% identified in the reference cohort (*p* < 0.00002). Among the patients with any germline variant, 62.5% (5 out of 8) reported a family history of cancer. Both cohorts identified variants in TP53 (*p* = 0.09004). Additionally, we observed variants in SERPINA1, CDKN2A, RAD50, and MUTYH, which were not identified in Huang’s study, even though these genes were interrogated [15] (Figure 2B).

Whereas in our cohort, 12% (3 out of 25) of our patients have variants in SERPINA1 there are no reports of variants in this gene in the reference cohort. This absence in the reference cohort is statistically significant (*p* = 8.66 × 10^−5^), suggesting this gene’s potential relevance for our Mexican patients. Although this variant has been reported to be more prevalent in Latin American populations, further investigation is required to assess its potential clinical significance. Overall, these results show a significant bias in the proportion of patients with known pathogenic germline variants in our cohort compared to the reference cohort, as well as variants not previously found in the Huang study (Figure 2B).

### 2.3. Somatic Mutations

#### 2.3.1. Mutational Signatures

Single base substitutions (SBS) refer to the replacement of a single nucleotide, taking into account not only the mutated base but also the immediate 5′ and 3′ flanking bases. Each signature displays a unique distribution pattern of these combinations. These somatic mutation patterns have been linked to distinct carcinogenic processes—such as tobacco exposure or UV radiation—and have recently emerged as promising biomarkers for cancer and predictors of treatment response [17].

We identified the contribution of a total of 36 single-base substitution (SBS) mutational signatures in our cohort, of which 29 are shared with the 64 signatures present in the reference TCGA-PanCancer cohort (syn1759475) (Figure 3A). To control for the effect of tumor mutational burden (TMB) differences, we used the mean proportions for each signature per individual. Signatures SBS40c (Unknown, suggested aging), SBS41 (Unknown, suggested colibactin), SBS59 (Artifact), and SBS85 (Cytidine deaminase) were only observed in our cohort, being absent in the reference cohort. When comparing the contributions of signatures shared between the two cohorts, the largest differences were found in SBS32, SBS5, SBS85, SBS12, and SBS95 (Bonferroni adjusted *p*-value < 0.0001).

The SBS5 signature is a clock-like signature, associated with aging, and is widely represented in all types of adult cancers. In our cohort the representation of SBS5 is higher compared to the reference cohort, even though the median age in our sample is lower (60 vs. 67 years, *p*-value = 0.007) [18]. Signature SBS32 is associated with immunosuppressive treatment with Azathioprine, and SBS85 is related to indirect effects of induced activation of cytidine deaminase. The etiology of the SBS12 signature is unknown, and SBS95 is associated with sequencing artifacts (Figure 3B). Most notably, the SBS4 signature, present in 67.6% of smokers in the reference cohort, was absent in our cohort, even in patients with a history of smoking (*p*-value = 0.01652) (Figure 3C). Taken together, these results show statistically significant differences in the somatic mutational signature profiles of our cohort when compared to a reference cohort originally enriched in European ancestry.

#### 2.3.2. Genes with Potential Positive Selection (Drivers)

A total of 22,927 single nucleotide variants (SNVs) and 2044 insertions and deletions (indels) were identified with a median TMB of 6.35 somatic SNVs per megabase. In order to identify mutated genes with possible positive selection in tumors (driver candidates), we used a combination of three methods (dNdScv, MutSigCV, and Oncodrivefml). These three tools are commonly used to identify genes under positive selection in tumors, each employing a slightly different methodology. Essentially, the higher a gene ranks, the more likely it is to be under positive selection in the tumor, and therefore, the greater its likelihood of being a driver gene.

However, their reliability in small samples may be compromised by potential random variation. To overcome this limitation, we incorporated bootstrapping and adopted a gene population-based approach to enhance statistical robustness. We used these tools to look for significant coincidences among the top 10% most highly ranked genes by all three methods when applied to both. First, we asked if a significant coincidence is observed in the gene ranking generated by all three tools when compared with chance expectations in our cohort, as assessed using a bootstrap analysis consisting of 50,000 equally sized random gene samples (see methods). This analysis showed a highly statistically significant coincidence for the top 10% ranking when comparing all three methods despite the small number of patients.

We next compared the 10% most highly ranked genes jointly detected by at least two of all three tools with those of the TCGA-PanCancer reference cohort (syn1759475), and found an overlap of 49 genes out of 631, a number of genes significantly higher than expected by chance (*p*-value < 2 × 10^−5^, as numerically assessed using 50,000 random samples of equal size drawn from each cohort and looking at the intersection) (Figure 4A).

We next assessed potential functional associations among the ranking 631 genes in our cohort by assessing the frequency of protein–protein interactions among the associated gene products. This analysis was conducted to ensure that the ranking does not prioritize random genes. It showed an enrichment of interactions among these genes above chance expectations (*p* = 0.0379). These results demonstrate that the top most highly ranked genes are statistically enriched in genes that are also detected in the top ranking genes in the reference cohort, and that the associated protein products of these genes engage in a number of protein–protein interactions higher than expected by chance, suggesting an effective functional association between these genes, possibly linked to tumor progression, in our cohort of Mexican patients (Figure 4B,C).

We wanted to compare the gene ranking of the reference data generated by our workflow and the ranking reported by the authors using their own workflow (syn1661643). To this end, we selected the top 10% of genes ranked by each approach and calculated the intersection between the resulting gene sets. As shown in Figure 5A, 40% of genes detected are not at the top in the PanCancer reference list. In our cohort, 88% of detected genes also fall outside of this top (Figure 5B). This result suggests the existence of a large number of potential drivers not previously identified in populations with European ancestry.

To confirm that this result was not an artifact of the small sample size of our cohort, we conducted a bootstrap analysis using random samples of 25 individuals drawn from the whole set of samples from the Pancancer cohort and obtained the expected distribution of intersections between them and the PanCancer reference list. As shown in Figure 5C, there is an effect of the sample size but the proportion of non-PanCancer genes among the top 10% was significantly higher than expected, demonstrating that the bias observed in our cohort is not an artifact of the reduced sample size of our cohort.

From this point, we divided the genes with the highest potential for positive selection or potential drivers (from the intersection of the first deciles) into those shared with the TCGA-PanCancer reference cohort and those not shared, which appear to be unique to our cohort.

The genes that showed the highest number of somatic mutations that are also present in the reference cohort are NRCAM (24%), EGFR (24%), TET1 (20%). However, the genes that have significantly different mutation frequency in our cohort compared to the reference cohort are SLC36A4 (*p* < 0.00002), AP1S1 (*p* < 0.00002), and TP53 (*p* = 0.00005). The mutation frequencies in EGFR (24% vs. 30%, *p* = 0.6572) and KRAS (16% vs. 10%, *p* = 0.3101), on the other hand, are similar to those reported in previous gene panel studies in the Mexican patients.

Oncoplots shown in Figure 6 correspond to either the top 25 most highly ranked genes using all three driver detection tools shared between our cohort and the reference cohort (Figure 6A), or to the 25 highest-ranked genes that are unique to our cohort (Figure 6B). Functional overrepresentation analyses were performed.

Among the top 25 genes shared within the Pan-cancer cohort, a significant enrichment of non-small cells lung cancer was observed, as expected. Although the pathway labels refer to different cancer types, the genes contributing to these enrichments are largely shared and have been described as playing important roles across multiple cancer types. As for the genes with the highest potential positive selection that were not shared with the reference cohort, RBMX (24%), PDE4D (20%), JAK2 (16%) were the most frequently mutated (Figure 6A). Among the top 25 genes unique to our cohort, the Necroptosis pathway was found significantly enriched (FDR = 0.0266), suggesting a potential functional role of this pathway. The mutated genes in our cohort contributing to this pathway were JAK2 (16%), CHMP2B (8%), MAPK8 (8%), and IFNA6 (8%) (Figure 6B). Taken together, our results reveal distinct genomic biases in lung cancer carcinogenesis in this Mexican cohort, when compared to a European ancestry reference population.

## 3. Discussion

Previous research has indicated that genomic variations among populations may reflect distinct carcinogenic processes in lung adenocarcinomas, which could influence precision medicine strategies when considering different communities.

It has been suggested that there is a genetic factor associated with Hispanic or Native American ancestry that influences the prognosis of these patients [19,20]. In a germline variants analysis conducted in our cohort of Mexican patients, we observed a pathogenic variant in TP53, also affected in the TCGA-PanCancer adenocarcinoma cohort [15]. TP53 is associated with Li-Fraumeni syndrome, which has been described as a significant risk factor for many types of cancer, particularly in young patients. Germline variants in RAD50 have been reported in patients with lung adenocarcinoma, but they have not been linked to an increased risk of LUAD [21]. Loss of function in *CDKN2A* has been associated with resistance to immunotherapy in patients with NSCLC [22]. Similarly, germline variants in *MUTYH* have been linked to an increased risk of lung cancer, particularly in non-smokers [23]. EGFR is the gene most strongly associated with susceptibility to lung cancer, yet we do not find such variants in our cohort [24,25,26,27]. On the other hand, SERPINA1, which encodes for alpha-1 antitrypsin, is affected in three of the patients in our cohort. Deficiency of this protein is strongly associated with the development of lung emphysema and chronic obstructive pulmonary disease. As a result, patients carrying pathogenic variants in this gene have an increased risk of lung cancer, especially adenocarcinoma subtype [24,28,29,30,31]. It has been proposed that one mechanism associated with increased LUAD risk is that alpha-1 antitrypsin deficiency increases the contact with agents in airways, leading to greater exposure to environmental carcinogens [32]. There is evidence for a link between this variant and tobacco or exposure to wood smoke in Mexicans with respiratory diseases, although this has yet to be studied in LUAD patients [33]. However, there are reports indicating that variants in this gene lead to a poorer response to chemotherapy [32]. Although this variant has been reported to be more prevalent in Latin American populations, further investigation is required to assess its potential clinical significance.

Patients in our cohort exhibit a higher frequency of pathogenic or likely pathogenic germline variants in cancer susceptibility genes (32% vs. 6.4%), although pathogenic variants have been reported in up to 18.5% of patients of European ancestry [34]. This could be due to the lack of clinical information to better discriminate each variant. However, this difference could also be influenced by technical factors, different methodologies, differences in variant classification criteria, and the fact that global reference databases are predominantly composed of individuals of European ancestry. As a result, variants that are common in Latin American populations may be erroneously classified. Further studies and a more comprehensive characterization of germline variants in Latin American populations are needed to explore the relevance and potential involvement of variants in SERPINA1 and other genes in LUAD among Mexican patients.

Smoking is the primary risk factor for lung adenocarcinoma globally. However, in developing countries, additional risk factors such as exposure to wood smoke and biofuels gain significance [35,36,37]. A total of 40% of the patients in our cohort had a history of smoking, which is lower than the reported proportion of patients of European ancestry (approximately 85%) [38]. Smoking status has also been associated with mutations in KRAS. Of the four patients with KRAS mutations, three have a history of smoking, and the smoking status of the remaining patient is unknown. Exposure to wood smoke is reported in 30–40% of lung adenocarcinoma cases, and this percentage increases in rural areas. In our cohort, we observed it in 22% (4 out of 18) of the patients. This risk factor has been linked to mutations in EGFR, adenocarcinoma histology, non-smokers, and better prognosis. No specific pattern was observed in the analysis of mutational signatures that could be related to wood smoke exposure.

Interestingly, none of the smoking patients from our cohort carried the SBS4 signature which is known for its robust association with this etiology [39,40]. Two independent studies on Asian patients with LUAD have demonstrated that the impact of this signature in smokers is less pronounced compared to individuals of European ancestry [41,42]. In a large Chinese cohort, the correlation with smoking status was not observed [42]. These findings support the hypothesis that carcinogenic processes are influenced by population-specific biases. Ancestry-related genetic variants in genes involved in carcinogen metabolism or DNA repair mechanisms may affect the processing of carcinogens or alter DNA topology, resulting in distinct mutagenic mechanisms and biases in mutational patterns. Another important consideration is the potential influence of germline variants on individual susceptibility to carcinogens, as well as the presence of other factors, such as exposure to uncharacterized environmental agents, that may mask the traditional effects of tobacco.

We did not observe well-known lung cancer signatures, such as APOBEC (SBS13), or they appeared with very low contributions (SBS2) despite being related with non- or light-smokers. A total of 66% of patients with EGFR mutations have a contribution from the SBS2 signature [39,40]. The mutational signatures with the most statistically significant differences in our cohort are SBS32, SBS85, SBS12, SBS19, and SBS95. The SBS32 signature has been associated with immunosuppressive treatment with Azathioprine. Although there are reports of its use in the treatment of interstitial lung disease, we lack information regarding exposure to this compound among our patients [43]. The SBS85 signature is associated with the indirect effect of induced activation of cytidine deaminase (AID), an enzyme homologous to APOBEC implicated in the development and progression of cancer. This signature is associated with mutations in TP53, which we observed in 2 out of 3 patients with this signature [44]. Somatic mutation-related features have been associated with response to immunotherapy, with tumor mutational burden (TMB) being the most common [45,46,47]. Additionally, hypermutation phenomena such as SBS32 and SBS85 signatures have also been considered. Moreover, some studies suggest that differences in the immune microenvironment may exist between populations [48,49]. Signature SBS32 has been associated with improved overall OS in ROS1-positive patients undergoing immunotherapy, although the underlying mechanism remains unclear [50]. In the case of SBS85, it has been hypothesized that the better prognosis observed in patients treated with immune checkpoint inhibitors (ICIs) may result from somatic hypermutation driven by AID activity which generates neoepitopes that enhance immunogenicity and leads to T-cell exhaustion—factors that could increase sensitivity to ICIs [50,51]. The SBS95 signature was originally described as an artifact; however, recent reports have indicated an association with the deamination process, similar to the SBS85 signature [52]. The SBS12 and SBS19 signatures are classified as of unknown etiology that show strain biases, suggesting that they are the product of the effect of exogenous agents. Associations with particular chemicals have been suggested [18,53,54]. The mutational signatures SBS19 and SBS5 have been linked to carcinogenic substances from environmental pollution, with SBS5 recently associated not only with aging but also with exposure to PM2.5 particles [55]. Additionally, the SBS5 and SBS12 signatures, along with SBS4, exhibit population-specific biases in Asian patients. These biases in mutational signature analysis highlight potential differences in the mechanisms of LUAD initiation and progression driven by environmental carcinogens. Such findings are particularly important as they can inform targeted health policies and diagnostic strategies for this specific population.

The genes with the highest potential for positive selection that are not shared with the Pan-cancer cohort were found in the top decile; there was an enrichment in protein–protein interactions greater than expected by chance, suggesting functional interactions among these genes, which may be likely involved in the carcinogenic process specific to our cohort. Among the top 25 genes most highly ranked as potential drivers, the necroptosis pathway was significantly enriched. This pathway is not enriched when evaluating the reference cohort or even the gene sets from cohorts with other ancestries. This cellular process is a poorly understood regulated cell death mechanism that, unlike apoptosis, involves an inflammatory response. Some studies in NSCLC patients have suggested that this pathway plays an important role in both tumor progression and antitumor processes through the regulation of the immune system. It can activate the adaptive immune response and trigger inflammation, or alternatively, contribute to the formation of an immunosuppressive microenvironment [56]. Targeted therapies have also been proposed for this pathway as they amplify anti-tumor immunity when combined with other treatments. Several genes within this pathway, including JAK2, IFNA6, and MAPK8, have been identified as drug targets. In addition, it has been suggested that high expression of key genes in this pathway is associated with improved progression-free survival, disease-free survival, and better response to chemotherapy and radiotherapy. However, there are also reports linking overexpression of these genes with a worse prognosis, which could be explained by excessive necroptosis-mediated damage. This pathway could be used to predict prognosis after treatment, and necroptosis-based anti-tumor therapies could be promising therapeutic strategies [57,58,59]. However, further work is needed to properly ascertain their clinical application.

The genes with the most significant differences in frequencies, when compared to the reference cohort, are SLC36A4, AP1S1, and TP53. The SLC36A4 gene belongs to a family of amino acid transporters that have been poorly studied in cancer; however, it is well known that amino acid transport is essential for the metabolism of cancer cells. Its critical role in migration is independent of its involvement in migration. It has been shown to play a role in cell growth regulation as well as in migration, generation of a immunosuppressive microenvironment and autophagy [60,61]. However, AP1S1 is a key component of cellular trafficking and receptor regulation, controlling the availability of surface molecules like growth factor receptors and extracellular matrix elements, reducing immunogenicity in cancer cells. Also, it has been associated with increased susceptibility to TKIs in patients with EGFR mutations due to EGFR lysosomal degradation [62]. Its fundamental role in copper homeostasis, which is important for carcinogenesis, has also been recognized [63]. On the other hand, TP53 plays a fundamental role in lung adenocarcinoma and ethnic differences have even been reported and proposed as prognostic markers for immunotherapy [64,65,66]. EGFR and KRAS are clinically relevant genes due to the availability of targeted therapies. In this study, we confirm the frequencies of somatic mutations in these genes in the Mexican population, with 24% for EGFR and 16% for KRAS, compared to 20% and 10%, respectively in the Carrot-Zhang recent study that included Mexican patients with lung adenocarcinoma [19]. The Hispanic paradox posits that Hispanic patients have better prognosis, at all stages, than European ancestry patients even in the presence of more adverse conditions, such as lower socioeconomic status [67]. A recent study suggests an association of EGFR mutations with Native American ancestry [19]; however, in our study these patients do not have a significantly higher contribution of Mexican indigenous ancestry. This discrepancy may be due to differences in Native American ancestry markers used in the cited study, or because the Carrot-Zhang study was conducted using tumor-only data.

Among the genes with high mutation frequencies in our cohort, we identified several potential biomarkers or targets of clinical interest (Appendix A) [68,69,70,71,72,73,74,75,76,77,78,79,80,81,82,83,84,85,86,87,88,89,90,91,92]. One notable example is TET1, which encodes an enzyme involved in DNA demethylation and is generally regarded as a tumor suppressor. However, under hypoxic conditions, TET1 may assume an oncogenic role. Mutations in this gene have been proposed as biomarkers for predicting the efficacy of immunotherapy and tyrosine kinase inhibitors (TKIs), such as Erlotinib, a commonly used targeted therapy for lung adenocarcinoma [93,94]. Furthermore, the co-occurrence of mutations in KRAS and TET1 has been associated with poorer survival outcomes, underscoring their clinical significance [95].

We also identified other genes, such as RBMX and PDE4D, which were present in our cohort but absent in the reference cohort and have previously been suggested as biomarkers. RBMX is a multifunctional RNA-binding protein involved in various processes, including RNA splicing, cell cycle progression, DNA repair, and metastasis. Notably, RBMX exhibited the highest number of mutations in our cohort. This gene has been linked to poor survival in bladder cancer, and tobacco-induced mutations in RBMX are thought to predispose individuals to lung cancer [96]. Additionally, both RBMX and PDE4D have been proposed as potential predictors of response to immunotherapy in patients with lung adenocarcinoma [88,97,98]. Moreover, some of these genes have been considered therapeutic targets, either alone or due to their synergistic effects when blocked in combination with existing therapies. In the case of RBMX, it has been proposed as a potential therapeutic target in lung cancer due to its association with telomere instability, and its depletion has been shown to sensitize tumor cells to cytotoxic agents. Similarly, the inhibition of AP1S1 has shown promising results in cells resistant to anti-EGFR TKIs [62]. Furthermore, PDE4D has been suggested as a potential therapeutic target in lung cancer as its silencing affects both proliferation and metastasis in tumor cells [99,100].

This initial study aims to identify genetic variations relevant to Mexican patients, though determining clinical relevance at the gene level remains challenging. One of the limitations of the sample size is that we could miss rare mutations in the somatic variants analysis. This issue could be further investigated in the next phase of the study, using a larger cohort. Despite this, we observe biases that suggest significant differences across various levels (germline, somatic signatures, and potential drivers) which warrant further exploration. Confirming these findings in a larger cohort and conducting functional assays are essential for a more thorough characterization of these differences. Our results have potential implications for precision medicine strategies tailored to Latin American populations. Addressing the current limitations and continuing this approach will bring us closer to developing personalized medicine that accounts for the unique genetic characteristics of populations like ours.

## 4. Materials and Methods

### 4.1. Mexican Lung Adenocarcinoma Cohort

Mexican patients treated at the Instituto Nacional de Enfermedades Respiratorias (INER) in Mexico City were included. The inclusion criteria were as follows: 1—confirmed lung adenocarcinoma by histopathological report, 2—smokers and non-smokers, 3—any age, 4—any stage, 5—signed informed consent. The project was approved by Instituto Nacional de Medicina Genomica’s Ethical and Research committees (C1_12/2010).

### 4.2. Sample Processing and Sequencing

DNA extraction was performed from frozen or paraffin-embedded lung tumor tissue and, for the ones available, peripheral blood or adjacent lung tissue as a paired sample. DNA extraction from these samples was carried out using Qiagen kits: QIAamp DNA Mini Kit QIAmp^®^ (Qiagen, Hilden, Germany, Cat. No: 51304), DNA FFPE Tissue^®^ (Qiagen, Hilden, Germany, Cat. No: 56404), and y QIAamp DNA Blood^®^ (Qiagen, Hilden, Germany, Cat. No: 51104), according to the manufacturer’s instructions. Library preparation and whole-exome sequencing were conducted through three external services: BGI Americas (Cambridge, MA, USA), INMEGEN, and Broad Institute, using Illumina platforms (San Diego, CA, USA), with paired-end sequencing.

### 4.3. Preprocessing

The quality of the raw reads was assessed using FastQC (v.0.11.9) and aligned to the hg19 reference genome (build GRCh37) obtained from the Broad Institute bioinformatics resource base using BWA (0.7.12-r1039) [101]. Preprocessing of the raw files was performed according to GATK best practices [102].

### 4.4. Ancestry Analysis

The ancestry inference was performed using ADMIXTURE (v.1.3.0) [103], which included 145,815 SNP’s and assumed three clusters (K = 3) for the calculation of the proportion of African, European and Indigenous Mexican. The references were derived from African (Yoruba), northern European (Utah), and indigenous Mexican from individuals included in the García-Ortiz’s study [12].

### 4.5. Germline Variants Analysis

The HaplotypeCaller (GATK 4.0.5.1) algorithm was run in joint calling mode according to best practices. We used GenotypeGVCFs, VariantRecalibrator, and ApplyVQSR using reference resources from: HapMap (3.3), 1000 Genomes (phase 1 and omni 2.5), and (dbsnp138). We filtered snps using: “QD < 2.0”, “FS > 60.0”, “MQ < 30.0”, “MQRankSum and “ReadPosRankSum < −8.0”. We focused on variants that: 1. were reported as “Pathogenic” or “Likely pathogenic”; 2. met ACMG and AMP recommendations [16]; 3. had an allele frequency of less than 0.05 in any population according to gnomAD and/or 1000 Genomes Phase 3; 4. belonged to the set of 152 cancer susceptibility genes [15], and 5. passed manual review using IGV (v.2.14.1) [104]. The variants of “Uncertain Significance” were selected after passing additionally two of three criteria: CADD (v.1.6) phred score higher than 20, classified as “Probably damaging” or “Possibly damaging” by Polyphen (v.2.2.2) or “Deleterious” by SIFT (v.5.2.2). Predictions were run through the Variant Effect Predictor (v110) [105].

### 4.6. Somatic Variants

Somatic variant identification was performed using Mutect2 (GATK 4.0.5.1), employing a combined panel of normal samples composed of non-tumor tissue samples from this cohort, a set of normal samples from the Mexican population used in a previous study by our group [106], and the panel of normal samples provided by the Broad Institute bioinformatics resource base. Variant annotation was performed using VEP (v110), followed by a manual review with IGV (v.2.14.1). The TMB was calculated on the basis of non-synonymous mutations (splice site, missense, nonsense, and nonstop mutations) divided by 37.5 Mb of the exome library used.

#### 4.6.1. Mutational Signatures Analysis

The extraction of mutational signatures was performed using SigProfilerMatrixGenerator (v.1.2.17) [107] and SigProfilerAssignment (v.0.1.0) [108] on the entire set of somatic variants after quality filtering. Subsequently, for cohort comparisons, we aim to mitigate biases related to TMB by calculating the mean individual contribution (as a proportion) to determine the contribution of each signature to the cohort.

#### 4.6.2. Positive Selection Analysis

To identify genes under positive selection in our cohort, we used dNdScv (v.0.0.1.0) [109], MutsigCV (v.1.3.5) [110], and Oncodrivefml (v.2.2.0) [111]. Due to the lack of statistical significance resulting from sample size, we designed a strategy with numerical tests (using bootstrapping) to find out if the rankings are randomly ordered by taking deciles of results from each tool and using the ranking of significant genes (syn1661643) as a control. Subsequently, we continued the analysis by considering the top decile and searching for intersections between tools and cohorts: ours and TCGA-PanCancer (syn1759475) run under the same parameters. We keep the genes that were in the top decile in two out of the three tools used. Overrepresentation analyses were performed using WebGestalt (v.151) [112]. Protein–protein interaction analyses were performed using STRING (v.12) [113] focusing on physical protein–protein interactions.

### 4.7. Statistical Analyses

Statistical analyses were conducted using R software (v.4.2.1). All comparisons and *p*-value calculations were performed using numerical tests, Wilcoxon–Mann–Whitney or Fisher tests with significance thresholds of *p* < 0.05 or lower, as specified in each case. Adjusted *p*-values in the analysis of mutational signatures were corrected for multiple testing using the Bonferroni method.

More detailed information can be found in the GitHub repository:

https://github.com/bgrueda/LUAD_MX_WES (accessed on 14 May 2025).

## Figures and Tables

**Figure 2 ijms-26-04865-f002:**
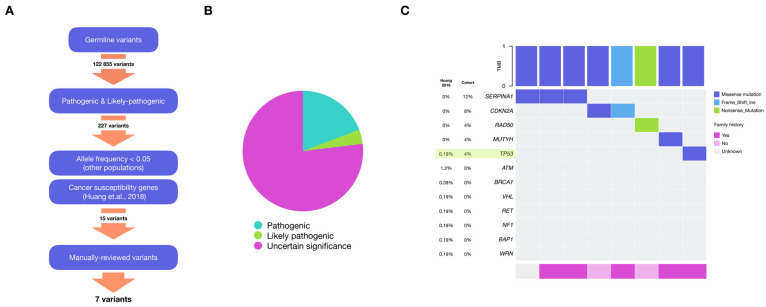
Pathogenic and Likely pathogenic germline variants. (**A**) Workflow of pathogenic and likely pathogenic germline short variants discovery. (**B**) Distribution of Uncertain significance, pathogenic and likely pathogenic germline variants (**C**) Details of genes affected by pathogenic and likely pathogenic germline variants in our cohort and the contrast with results reported by (Huang et al., 2018 [15]). In our cohort, 32% of patients exhibit germline variants, compared to only 6.4% in the reference cohort. In both cohorts, variants in TP53 were present (*p*-value = 0.09004), whereas SERPINA1 is the most affected gene in our cohort and was not reported to be affected in the reference cohort (*p*-value = 8.66 × 10^−5^).

**Figure 3 ijms-26-04865-f003:**
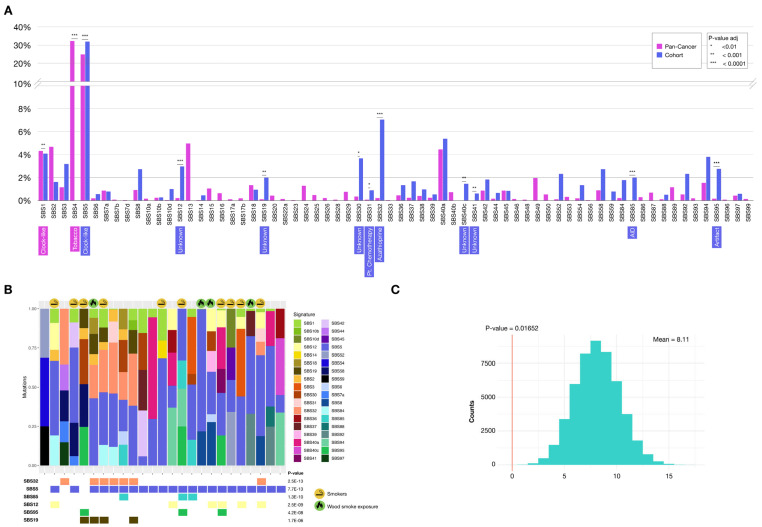
Mutational signature patterns. (**A**) Comparison of mutational signature profiles between our cohort and the TCGA-PanCancer cohort (syn1759475). (**B**) Proportion of mutational signatures per patient. (**C**) The SBS4 signature, associated with tobacco, does not appear in the cohort, even in patients with a history of smoking, and this absence was statistically significant (inset red vertical line).

**Figure 4 ijms-26-04865-f004:**
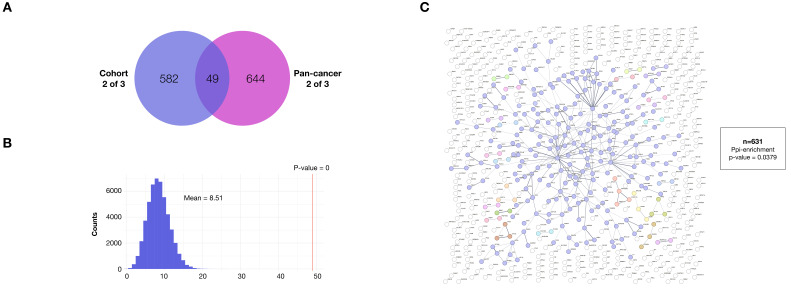
Enrichment of protein–protein interactions among genes most highly ranked for potential positive selection using three independent driver gene tools. (**A**) Enrichment of significant driver genes identified in the TCGA-PanCancer reference cohort (syn1759475), within the first decile of genes with most potential positive selection in our cohort. (**B**) Histogram showing the numerically assessed (bootstrap) expected intersection between the genes in the top decile of the PanCancer reference cohort and the top ranking (potential drivers) in our cohort. Actual intersections indicated (red line). (**C**) Protein–protein interaction network of the top-ranking genes in the cohort of Mexican patients (*p*-value = 0.0379).

**Figure 5 ijms-26-04865-f005:**
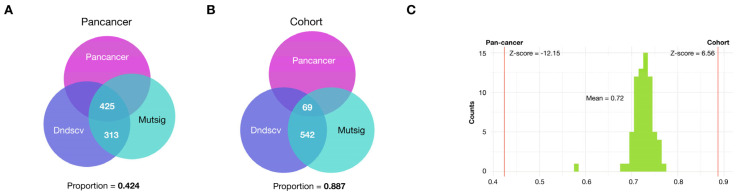
The small sample size does not explain observed bias in potential drivers. (**A**) Genes jointly called by Dndscv and Mutsig in the Pan-cancer cohort and their intersection with the Pan-cancer reference top 10% ranked list (syn1661643). (**B**) Genes jointly called by the two tools in our cohort and their intersection with the Pan-cancer reference top 10% ranked list (syn1661643). (**C**) A sample size effect is demonstrated by taking 25 random samples from the reference cohort and assessing the intersections of the first decile with a reported list of driver genes from the same cohort. The proportions of genes non-shared with the reference list is larger in our cohort, even when the size effect is considered.

**Figure 6 ijms-26-04865-f006:**
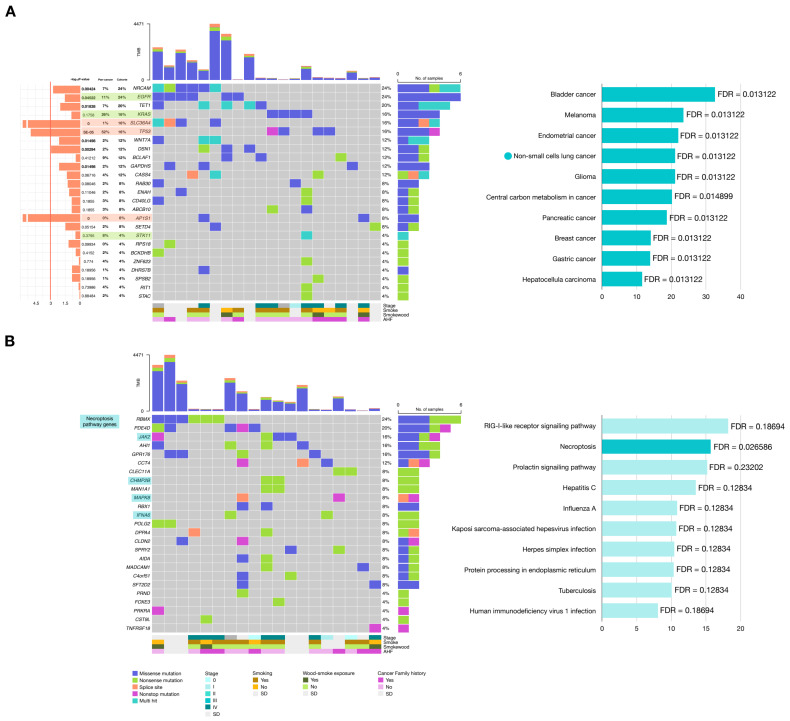
SNVs are among top ranking genes for their highest potential for positive selection. (**A**) Top 25 genes with the highest potential for positive selection shared with the TCGA-PanCancer reference cohort (syn1759475) and its corresponding pathway enrichment result. (**B**) Top 25 genes with the highest potential for positive selection specific to our cohort and its enriched pathways.

## Data Availability

Raw data for this study were generated at the Instituto Nacional de Medicina Genómica. The raw data supporting the conclusions of this article will be made available by the authors without undue reservation.

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
