# Peer review of "A Pilot Study: Contrasting Genomic Profiles of Lung Adenocarcinoma Between Patients of European and Latin American Ancestry"

_ijms, 2025, doi:10.3390/ijms26104865_

Round 1
Reviewer 1 Report
Comments and Suggestions for Authors
Comments:
- The manuscript lacks a direct comparison of clinical outcomes between Mexican and European lung cancer patients. A retrospective Cox proportional hazards model, adjusted for age, smoking, and tumor stage, is recommended to assess how ancestry-driven mutations affect prognosis.
- The study reports ancestry-specific genomic differences but lacks validation in an independent Latin American cohort. It would strengthen the results to include more Mexican lung cancer patients and validate findings using publicly available datasets (e.g., TCGA, ICGC) to compare genetic alterations across ancestries.
- The absence of the SBS4 signature in Mexican smokers is an intriguing finding but requires further validation. It’s better to analyze existing WGS data from Mexican patients stratified by smoking status to determine if SBS4 is truly absent or underrepresented.
- The comparison between smoking and wood smoke exposure in LUAD patients is valuable, but further clarity is needed on why SBS4 is absent in smoking patients, given its strong association with smoking in other populations.
- Expand on how mutational signatures, such as SBS32 and SBS85, could impact potential treatment strategies, including immunotherapies or TKIs, especially in the context of ancestry-specific genomic alterations.
- The study identifies a higher prevalence of germline pathogenic variants in Mexican patients but does not assess their functional relevance. Please discuss how these variants may influence cancer progression or treatment responses.
- The role of SLC36A4 and AP1S1 mutations in lung cancer progression is not well-explained. Consider performing pathway analysis (e.g., GSEA, KEGG) to determine the biological impact of these mutations.
- Necroptosis-related mutations are enriched in the Mexican cohort, but their role in lung cancer pathophysiology is not addressed. Please discuss this limitation.
- The study suggests ancestry-specific genomic alterations may impact treatment response but does not provide direct evidence. To strengthen this claim, Kaplan-Meier survival analysis with a log-rank test should be performed to assess the impact of ancestry-specific mutations on overall survival (OS) and progression-free survival (PFS) in an existing patient cohort. Multivariate Cox regression should also be conducted to adjust for clinical confounders.
- Discuss how immune microenvironment differences between Mexican and European patients support the findings.
- The study identifies RBMX and PDE4D as potential predictors of immunotherapy response but lacks clinical validation. Analyze patient survival data to assess their correlation with immunotherapy outcomes. Consider random forest or LASSO regression for therapy response prediction.
- Logistic regression modeling should be performed to assess the predictive value of ancestry-specific biomarkers for treatment selection, utilizing available patient cohorts.
- Please discuss how the genetic composition of the cohort (53% Indigenous Mexican, 42% European, 4.8% African) may impact the interpretation of genetic susceptibility and somatic mutations.
- The absence of the SBS4 mutational signature in smokers, despite a history of smoking, warrants deeper investigation.
- The use of multiple mutational signature tools (e.g., dNdScv, MutSigCV, Oncodrivefml) is robust, but the impact of small sample sizes should be acknowledged, especially given the variability of results between the Mexican and European cohorts.
- The oncoplot presentation (Figures 6A and 6B) is informative. However, a clearer explanation of the rationale behind selecting the top 25 genes and how they relate to potential driver genes in lung cancer would enhance the analysis.
- The finding of significant enrichment in the Necroptosis pathway is intriguing and novel. Consider comparing the enrichment of cancer-related pathways in this cohort with those identified in other populations to highlight unique vulnerabilities in the Mexican cohort.
- Clarify the methods for identifying germline variants and the criteria for selecting the 152 cancer-susceptibility genes.
- Include additional details on how tumor mutational burden (TMB) was calculated and its relevance to the study outcomes.
- Expand on the clinical significance of variants in genes like SERPINA1, which may be more prevalent in Latin American populations.
- It would be useful to provide additional information on the clinical significance or functional characterization of genes listed in Table S2, including any direct associations with survival outcomes or therapy responses.
Author Response
Thank you very much for your time and valuable contribution to this manuscript. We have attached a PDF with point-by-point responses, as well as an updated version of the manuscript that incorporates your comments and those from the other reviewers.

Reviewer 2 Report
Comments and Suggestions for Authors
Bertha et al. used genomic analysis to compare the profiles of lung adenocarcinoma of two populations of Mexican and European descent. I have the following concerns.
Major concerns
- In section 2.3, the authors define single base substitution mutational signatures like SB40c, SB41, and so on. It would be good to mention how 40c and 41 are defined. What do they mean?
- In sectional 2.3.2, the authors identified potential drivers. Are they specific to the Mexican population when compared to the European ? The authors showed genes with the highest number of somatic mutations, NRCAM, EGFR, and TETI. These genes are druggable?
- In Figure 6, A and B pathway enrichments are visualized. But they represent a variety of cancer-specific pathways. It might be confusing for readers as the study is focused on lung cancer. In Figure 6B, necroptosis is enriched. Authors need to highlight the genes from this pathway, which is specific to lung cancer in the Mexican population.
Author Response
Thank you very much for your time and valuable contribution to this manuscript. We have attached a PDF with point-by-point responses, as well as an updated version of the manuscript that incorporates your comments and those from the other reviewers

Reviewer 3 Report
Comments and Suggestions for Authors
Comments for authors:
This manuscript provides a valuable insights to the cancer genomics of underrepresented populations. The use of comparative analysis with TCGA-PanCancer cohorts is well-conceived and offers insight into ancestry-specific cancer biology.
Section 2.1: The patient details and sample numbers, though mentioned in Results & methods, it is not easy to understand while reading. The variable Denominators can still be explained. Example: Line 86-87: “The study included 11 males and 14 females, with the majority (9 out of 14, 64%) being at stage IV” it sounds like 9 females out of 14 females are being at stage 4, and there is no data about males. Including a statement “some data are missing/unavailable” before stating the proportions (in Line 88) will make it easier for readers to understand.
Section 2.2: The pathogenic variant frequency comparison i.e., 32% in yours and 6.4% in TCGA is convincing and well discussed. Yet the difference might also be influenced by technical factors including - differences in the gene panels analyzed, how variants are classified, and the fact that global databases like gnomAD are largely based on European populations. Because of this, variants common in Latin American or Indigenous populations may be misclassified as rare or pathogenic. Including this point would provide a more balanced explanation and highlight the importance of improving representation in genomic resources. Consider acknowledging this point.
Section 2.3.2 Authors mention that genes enriched in the cohort – JAK2/IFNA6 etc. are involved in the necroptosis pathway. This PPIs (derived from STRING) does not prove that these genes are functionally linked in the biological context. The interpretation can be clarified that “the results suggest potential functional relationships; enrichment was identified statistically via STRING” , The gene co-occurrence does not prove functional interaction without further experimental validation
Other minor corrections:
Rephrase line 101 : All patients in the cohort have indigenous Mexican ancestry [is that “all patients have some proportion of indigenous ancestry?]
Line 129 : Check abbreviation – AGMG – ACMG
Rephrase line 239 – please make it more clear
Expand AF in figure 2A
Comments on the Quality of English Language
Paragraphs can be split up especially in results section.
English can be edited for better understanding.
Author Response

(The authors gave the same response as above.)

Round 2
Reviewer 1 Report
Comments and Suggestions for Authors
Dear Authors,
Thank you for addressing my initial comments. The manuscript has improved following the revision. I have no further concerns.
Best,
Abhishek Tyagi